# GLUT1 Expression in Cutaneous Sebaceous Lesions Determined by Immunohistochemical Staining Patterns

Cynthia Reyes Barron [ID] and Bruce R. Smoller *[ID]

Department of Pathology and Laboratory Medicine, University of Rochester Medical Center, Rochester, NY 14642, USA; creyesba711@gmail.com
* Correspondence: bruce_smoller@urmc.rochester.edu

**Abstract:** GLUT1 is a membrane associated carrier protein that functions in the physiologic transport of glucose across cell membranes. Multiple studies have shown an increased GLUT1 expression in various tumor types and a role in cancer prognosis. The aim of this study was to determine whether cutaneous sebaceous lesions have a differential expression of GLUT1 by immunohistochemistry (IHC). GLUT1 IHC was performed on excision specimens of ten cases of sebaceous carcinoma, nine of sebaceoma, ten of sebaceous adenoma, and ten of sebaceous hyperplasia. Intense, diffuse cytoplasmic staining was observed in sebaceous carcinoma. The pattern of GLUT1 staining in sebaceomas and sebaceous adenomas consisted of a gradient of intense cytoplasmic staining in the basaloid cells with a decreased intensity to membranous staining only and absent staining in mature sebaceous cells. In lesions of sebaceous hyperplasia, GLUT1 staining outlined the basal layer of each gland; cytoplasmic staining was minimal to absent. Increased cytoplasmic staining of GLUT1 may correlate with cellular metabolic and proliferative activity. GLUT1 has potential utility in differentiating sebaceous lesions.

**Keywords:** GLUT1; sebaceous carcinoma; sebaceoma; sebaceous adenoma; sebaceous hyperplasia

## 1. Introduction

Sebaceous gland hyperplasia is commonly encountered in skin biopsy specimens while sebaceous neoplasms are relatively rare. An accurate diagnosis is imperative for treatment and follow-up, and in determining which patients should be tested for mismatch repair (MMR) deficiency, seen in Muir Torre syndrome [1]. This syndrome often presents initially with cutaneous sebaceous tumors, including sebaceous carcinoma, sebaceoma, and sebaceous adenoma. Unlike sebaceous neoplasms, sebaceous hyperplasia has no such association [2]. Sebaceous neoplasms may be clinically indistinguishable from other tumors, including basal cell and squamous cell carcinomas, requiring biopsy and histopathologic analysis for diagnosis. Benign lesions fall into a histologic spectrum with sebaceous adenomas consisting primarily of mature sebocytes with a minority of immature basaloid cells and sebaceomas displaying the opposite ratio of mature sebocytes to immature basaloid cells. An infiltrative growth pattern, cytologic atypia, prominent mitoses and necrosis are malignant features indicative of sebaceous carcinoma. Sebaceous carcinomas range from well-differentiated and clearly of sebaceous origin to poorly differentiated with few identifiable sebocytes [1]. The span of histologic findings make the diagnosis of sebaceous neoplasms challenging in many cases with high inter-observer variability [3].

Glucose, an essential source of energy for rapidly proliferating cells, is transported through highly regulated membrane-associated proteins of the GLUT family [4]. The GLUT1 transporter, found ubiquitously in cell membranes of tissues throughout the body, is prominent in keratinocytes of the basal layer of the epidermis in accordance with their proliferative nature [4]. The Warburg effect, first described by Otto Warburg in the 1920s, is the altered metabolism observed in tumor cells which results in rapid production of ATP with an increase in glucose uptake and lactate production [5]. Multiple studies have shown that the expression of GLUT1 in the cells of malignant tumors of various

types may be pathologically increased and this increased expression signals a worse prognosis [6]. GLUT1 expression increases as normal mucosal tissue becomes dysplastic and progresses to invasive carcinoma at oral sites [7]. In cervical mucosa, GLUT1 shows weaker expression in non-dysplastic cells and the expression increases in lesions that progress to high-grade dysplasia and carcinoma [8]. Increased expression is also correlated with increased proliferation and metastasis in melanoma as well as worse prognosis [9–11]. The purpose of this study was to explore possible differential expression throughout the spectrum of sebaceous neoplasms.

## 2. Materials and Methods

The laboratory information system of the surgical pathology department of the University of Rochester Medical Center (URMC) was queried to identify cases of sebaceous hyperplasia, sebaceous adenoma, sebaceoma, and sebaceous carcinoma diagnosed from 1 January 2010 to 1 January 2020. The slides were retrieved and reviewed to verify the diagnoses and cases with sufficient tissue for additional immunohistochemical (IHC) studies were selected from each category to include ten cases each of sebaceous hyperplasia, sebaceous adenoma, and sebaceous carcinoma; only nine cases of sebaceoma were available. The patient age, gender, and tumor site were noted. In addition, the mismatch repair (MMR) protein status was reviewed in cases where immunohistochemical studies for loss of these proteins were previously conducted in the URMC laboratory. The antibody clones used for MMR evaluation were: MLH1 (M1), MSH2 (G219-1129), MSH6 (SP93), and PMS2 (A16-4).

The tissue blocks for the selected cases were retrieved and slides cut for IHC staining with a monoclonal GLUT1 antibody (SPM498). The antibody is validated and currently in use for clinical cases. The results of GLUT1 IHC were evaluated using standard light microscopy.

## 3. Results

Thirty-nine sebaceous lesions were assessed. The average patient age was 66 years (range 40 to 89), 69% were male, and most lesions arose on the face or head and neck (see Table 1). Sebaceous carcinomas were more likely to be located at sites other than the face, including the trunk and proximal extremities.

**Table 1.** Features of cases studied according to diagnosis. Results of mismatch repair deficiency studies are listed where available.

| Diagnosis | Sebaceous Carcinoma | Sebaceoma | Sebaceous Adenoma | Sebaceous Hyperplasia |
|---|---|---|---|---|
| # of cases | 10 | 9 | 10 | 10 |
| Age (years) Average Range | 64 47–89 | 67 46–82 | 74 68–88 | 58 40–84 |
| Gender (male) | 60% | 56% | 90% | 70% |
| Sites | Eyelid, nose, face (other), scalp, neck, shoulder, chest, flank, thigh | Nose, ear, face (other), back | Nose, face (other), scalp, back | Nose, face (other), chest, thigh |
| GLUT1 Staining Pattern | Diffuse cytoplasmic and membranous staining in basaloid cells (variable) | >50% Diffuse cytoplasmic and membranous staining | <50% Diffuse cytoplasmic and membranous staining in greater than 1 layer | Only single layer of basaloid cells highlighted by GLUT1 |
| MMR [1] deficiency All retained MLH1/PMS2 MSH2/MSH6 MSH6 only not done | 1/10 1/10 6/10 1/10 1/10 | 3/9 0/9 1/9 0/9 5/9 | 6/10 0/10 2/10 0/10 2/10 | 10/10 |

[1] Mismatch repair deficiency previously assessed by protein loss using immunohistochemistry with antibodies for MLH1, PMS2, MSH2, and MSH6.

GLUT1 IHC staining was distinct between the various diagnostic categories. Sebaceous carcinomas displayed intense cytoplasmic and membranous staining that was diffuse throughout the basaloid cells (see Figure 1). Sebaceomas and sebaceous adenomas displayed a gradient with intense cytoplasmic and membranous staining in the basaloid cells with a decrease in intensity to membranous staining only to absent staining in mature sebaceous cells (see Figures 2 and 3). In lesions of sebaceous hyperplasia, GLUT1 staining highlighted the basal layer of each gland with minimal to absent cytoplasmic staining and completely absent staining in mature sebocytes (see Figure 4).

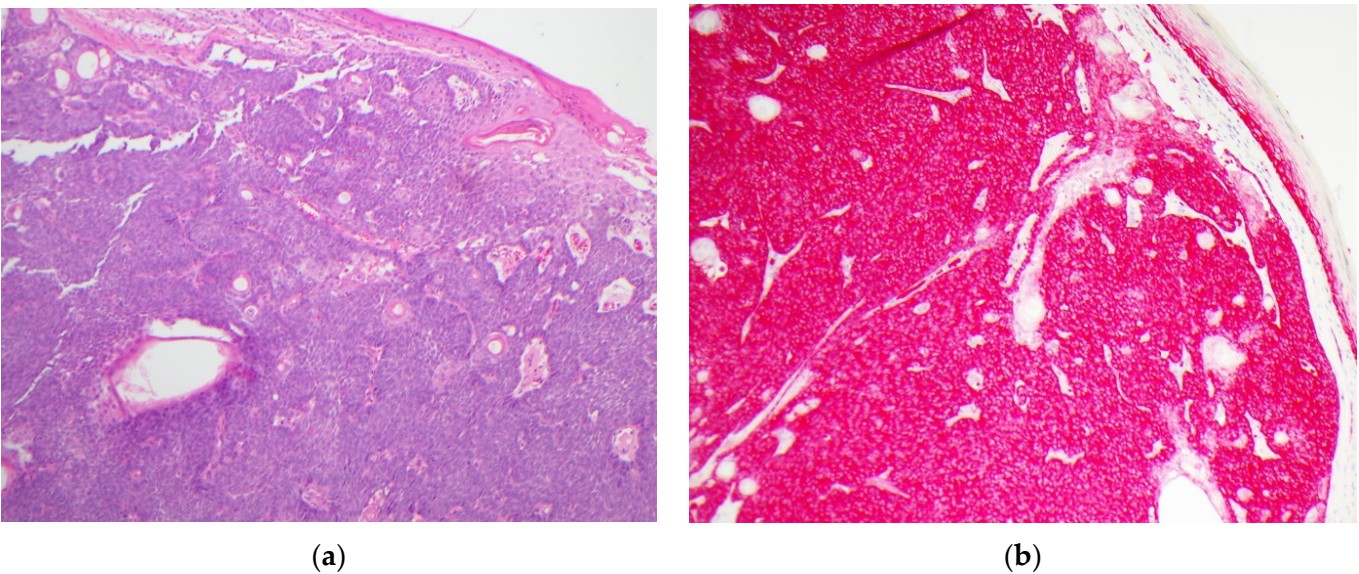

(**a**) (**b**)

**Figure 1.** Sebaceous carcinoma, sample case. (**a**) Hematoxylin and eosin-stained section showing an infiltrative proliferation of basaloid cells with focal sebaceous differentiation and abundant mitoses, original magnification 100×; (**b**) GLUT1 immunohistochemical stain of the same lesion showing strong diffuse cytoplasmic and membranous staining throughout the tumor, original magnification 100×.

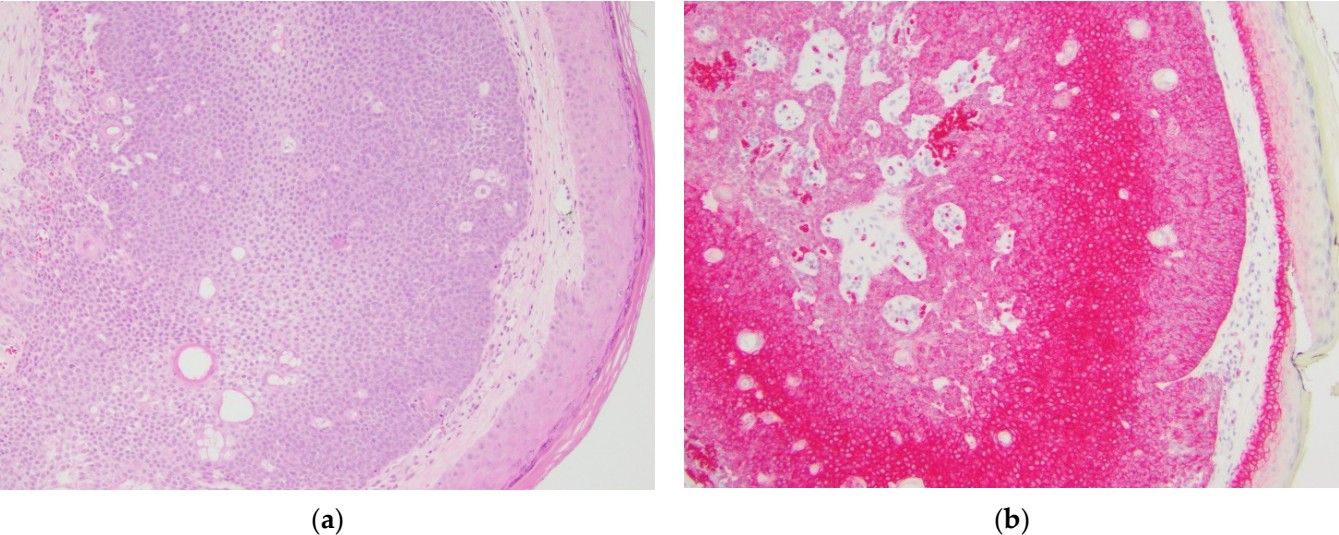

(**a**) (**b**)

**Figure 2.** Sebaceoma, sample case. (**a**) Hematoxylin and eosin-stained section showing a well-circumscribed dermal tumor consisting predominantly of basaloid cells with minimal cytologic atypia and mitoses, original magnification 100×; (**b**) GLUT1 immunohistochemical stain of the same tumor showing diffuse strong cytoplasmic and membranous staining in the basaloid cells with weaker staining in cells with greater maturation, original magnification 100×.

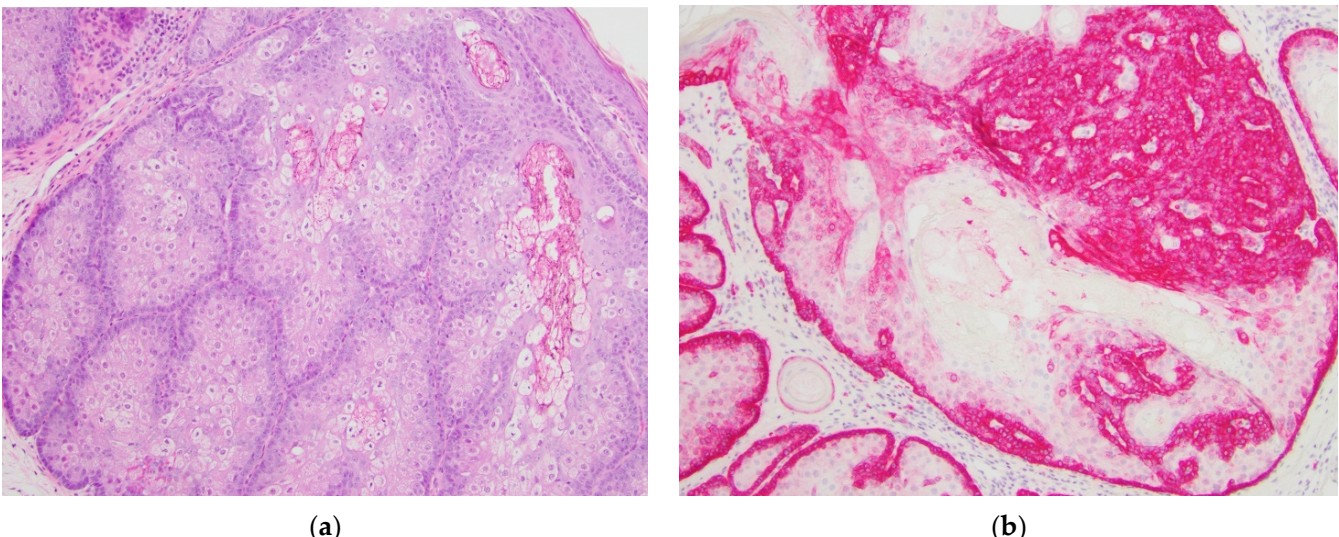

| (a) | (b) |

**Figure 3.** Sebaceous adenoma, sample case. (**a**) Hematoxylin and eosin-stained section showing a well-circumscribed dermal-based tumor consisting predominantly of mature sebocytes with a basaloid component comprising less than 50% of the tumor cells. Cytologic atypia and mitoses are not prominent, original magnification 100×; (**b**) GLUT1 immunohistochemical stain of the same tumor showing cytoplasmic and membranous staining in the basaloid component of the tumor with weak to absent staining in mature sebocytes, original magnification 100×.

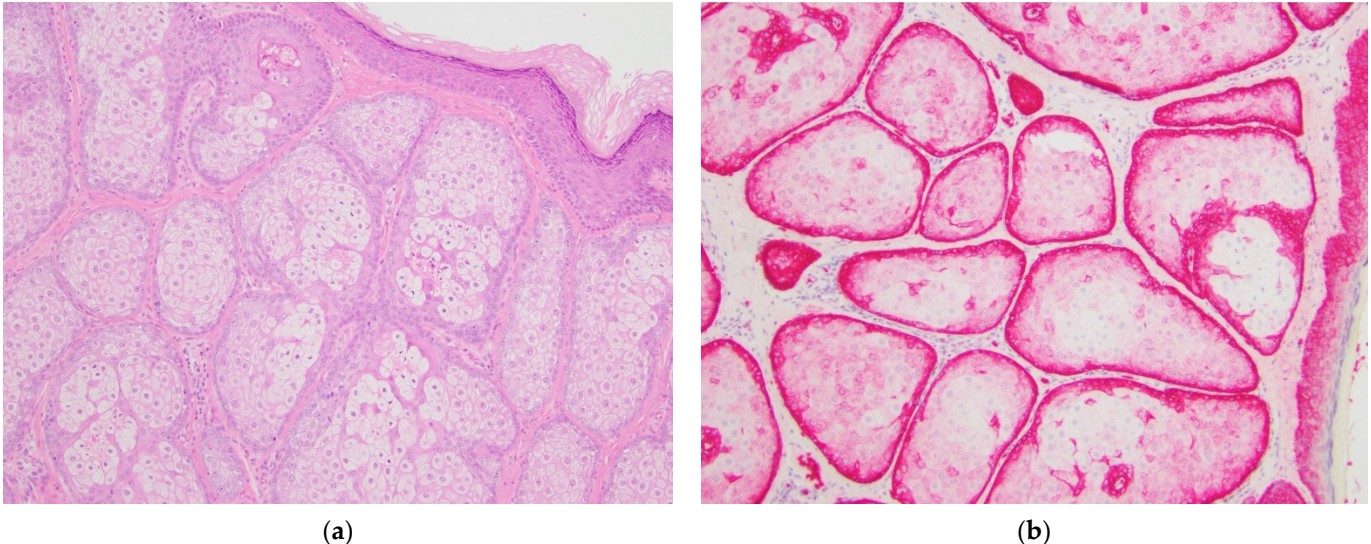

| (a) | (b) |

**Figure 4.** Sebaceous hyperplasia, sample case. (**a**) Hematoxylin and eosin-stained section showing a benign proliferation of sebaceous glands with a single layer of basaloid cells surrounding each lobule, original magnification 100×; (**b**) GLUT1 immunohistochemical stain of the same tumor showing primarily membranous staining only in the basaloid cells at the periphery of each lobule with staining becoming weaker to completely absent in the central mature sebocytes, original magnification 100×.

Results for mismatch repair deficiency were available for 21 of the 39 cases. Testing had been assessed using IHC staining for proteins MLH1, PMS2, MSH2, and MSH6. Cases with the loss of at least one of these proteins were considered deficient. In sebaceous carcinomas, testing was conducted in 9/10 cases and there was a deficiency in 8/10 cases with the loss of MSH2 and/or MSH6 seen in 6/10 cases. Testing was not performed on the majority of sebaceoma cases but one case was deficient and two sebaceous adenoma cases were also deficient; protein losses were identified for MSH2 and/or MSH6 in these lesions. No cases of sebaceous hyperplasia had undergone IHC testing for MMR deficiency (see

Table 1). The pattern of GLUT1 staining observed was no different between lesions with MMR protein retention or loss.

## 4. Discussion

Sebaceous adenomas, sebaceomas, and sebaceous carcinomas are neoplasms with variable proportions of immature and proliferating sebocytes. In contrast, sebaceous hyperplasia is a benign process with normal maturation in which sebocyte turnover is slowed and glands appear more numerous and crowded [1]. Factor XIIIa effectively highlights sebocytes with strong nuclear staining, helpful in distinguishing sebaceous neoplasms from other cutaneous lesions that may be histologic mimics such as squamous cell carcinoma with clear cell change [12,13]. Additional immunohistochemical markers, including EMA, adipophilin, and androgen receptor, may be useful in discriminating sebaceous carcinoma from other non-melanoma skin cancers as part of a panel [14,15]. Androgen receptor IHC highlights nuclei in sebocytes more diffusely in benign lesions than in poorly differentiated sebaceous carcinomas and sebaceous carcinomas only express EMA in well differentiated cells [16]. Adipophilin may show the opposite staining pattern as GLUT1 because it highlights the lipid droplets in mature, well differentiated sebocytes and is negative in the basaloid cells [17]. The staining pattern of D2-40 in sebaceous lesions is more similar to GLUT1 because it highlights basaloid cells in benign sebaceous hyperplasia, sebaceous adenoma, and sebaceoma, but the staining pattern is weaker and patchy in sebaceous carcinomas, particularly those that are poorly differentiated [18]. The percentage of nuclear survivin expression may vary significantly between sebaceous lesions aiding in categorization [19]; however, this marker may not be available in many labs. No immunohistochemical marker is currently commonly used to classify lesions along the spectrum of differentiation of sebaceous neoplasms. GLUT1 may be a helpful marker in distinguishing between categories of benign sebaceous lesions and malignant sebaceous carcinoma and aid in categorization. As illustrated in Figure 4, GLUT1 staining is sharper and cleaner than standard H&E, providing a better distinction between basaloid cells and mature sebocytes. In cases where the differential diagnosis lies between sebaceous hyperplasia and sebaceous adenoma, GLUT1 IHC will clearly highlight immature basaloid cells. In Figure 3, portions of the lesion have more than one layer of basaloid cells, supporting the diagnosis of sebaceous adenoma over sebaceous hyperplasia. Again, in this example, the distinction is easier to make using GLUT1 IHC than standard H&E.

Testing for MMR deficiency by IHC in sebaceous neoplasms is common practice and an important step in the diagnosis of Muir-Torre syndrome [2,20]. As expected, MMR IHC had not been performed on cases of sebaceous hyperplasia included in this study, but had been performed on the majority of neoplasms. Deficiencies were identified in 52% of sebaceous neoplasm cases that were screened by IHC. MSH2 is the most frequently deficient gene in Muir-Torre syndrome with loss in up to 90% of cases [2]. Of the sebaceous carcinomas in this study, 67% had loss of the MSH2 expression by IHC. Notably, there was no significant difference in GLUT1 staining patterns between lesions in the same category that had loss of at least one MMR protein or retention of all proteins. The loss of the MMR protein expression should prompt consideration of genetic counseling for patients and family members as well as heightened cancer screening [2]. In most cases, the distinction between sebaceous hyperplasia and sebaceous carcinoma or sebaceoma is straightforward. The distinction may not be as clear in certain cases of sebaceous adenoma where the basal proliferation is minimal and the majority of the tumor consists of mature sebocytes. An accurate diagnosis in such cases is imperative when considering which tumors to test for MMR deficiencies due to the implications for Muir-Torre syndrome diagnosis. GLUT1 immunohistochemistry, with its sharp variable staining, may be a valuable tool for pathologists for such cases.

Sebaceous neoplasms and sebaceous hyperplasia display distinctive patterns of GLUT1 staining by immunohistochemistry. The patterns seen may correlate with cell immaturity and higher metabolic requirements for glucose in proliferating cells. Although

GLUT1 is a membrane protein, the strong cytoplasmic staining seen in neoplastic cells demonstrates the presence of the protein in high quantities beyond the membrane, within the cytoplasm. Increased cytoplasmic staining of GLUT1 may correlate with cellular metabolic and proliferative activity in poorly differentiated, immature sebocytes. Additional immunohistochemical studies with a GLUT1 antibody will be helpful in elucidating the role of this glucose transporter in sebaceous cutaneous lesions and may aid in an accurate diagnosis. Research is currently underway to determine the utility of GLUT1 inhibitors for targeted treatment of various malignancies with multiple clinical trials in progress [6]. In the skin, topical GLUT1 inhibitor therapy has shown promise for psoriasis treatment with the ability to decrease keratinocyte hyperplasia [4]. The possible implications of GLUT1 expression for prognosis and treatment in sebaceous neoplasms remains to be explored.

**Author Contributions:** All authors contributed to conceptualization, methodology, formal analysis and investigation. Writing of the original draft was performed by C.R.B. Supervision and project administration were conducted by B.R.S. All authors have read and agreed to the published version of the manuscript.

**Funding:** This research received no external funding.

**Institutional Review Board Statement:** Not applicable.

**Informed Consent Statement:** Not applicable.

**Conflicts of Interest:** The authors declare no conflict of interest.

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
