# Peer review of "GLUT1 Expression in Cutaneous Sebaceous Lesions Determined by Immunohistochemical Staining Patterns"

_dermatopathology, doi:10.3390/dermatopathology8030031_

Round 1

Reviewer 1 Report

This is an original study on GLUT1 expression in a range of sebaceous neoplasms. The study is not uninteresting but could be improved. Please consider the following:

  • the introduction is somewhat wordy and confusing; It shoud focus on the main topic investigated, ie the utility of GLUT1 in classifying/diagnosing sebaceous tumours. Too many data on Muit-Torre syndrome and MMR deficiency are in my opinion secondary and should only be put in perspective with the expression of GLUT1 as the leading aim of the study
  • the correlation between GLUT1 expression and MMR tests should be described in the results (it appears only in the discussion!)  
  • some form of quantification of the immunostaining results for GLUT1 could be provided, eg percentage of positive cells; this could be incorporated in table 1. Furthermore, as the authors hint that GLUT1 immunostaining can be of help to distinguish the various sebaceous tumors between them (and overall distinguish sebaceous hyperplasia from sebaceous adenoma for the  diagnosis of MTS), they could try to set cut-off values for each tumor category, otherwise the statement is too vague to be helpful in practice. 
  • it would be interesting to know how GLUT1 expression across the spectrum of sebaceous tumours compares with other known sebaceous antigens (eg EMA, androgen receptors, adipophilin); if the authors do not have personal data on this topic, they could compare their results on GLUT1 with results from the literature regarding EMA, AR, adipophilin.   

Author Response

Author responses to reviewer comments:

Comments and Suggestions for Authors

This is an original study on GLUT1 expression in a range of sebaceous neoplasms. The study is not uninteresting but could be improved. Please consider the following:

the introduction is somewhat wordy and confusing; It shoud focus on the main topic investigated, ie the utility of GLUT1 in classifying/diagnosing sebaceous tumours. Too many data on Muit-Torre syndrome and MMR deficiency are in my opinion secondary and should only be put in perspective with the expression of GLUT1 as the leading aim of the study

The introduction has been shortened and information on Muir-Torre syndrome has been deleted.

the correlation between GLUT1 expression and MMR tests should be described in the results (it appears only in the discussion!) 

A sentence has been added to the last paragraph of the results explaining that there is no correlation.

some form of quantification of the immunostaining results for GLUT1 could be provided, eg percentage of positive cells; this could be incorporated in table 1. Furthermore, as the authors hint that GLUT1 immunostaining can be of help to distinguish the various sebaceous tumors between them (and overall distinguish sebaceous hyperplasia from sebaceous adenoma for the  diagnosis of MTS), they could try to set cut-off values for each tumor category, otherwise the statement is too vague to be helpful in practice.

A row has been added to Table 1 to summarize the GLUT1 findings in the various lesions and additional explanation provided at the end of the first paragraph of the discussion.

it would be interesting to know how GLUT1 expression across the spectrum of sebaceous tumours compares with other known sebaceous antigens (eg EMA, androgen receptors, adipophilin); if the authors do not have personal data on this topic, they could compare their results on GLUT1 with results from the literature regarding EMA, AR, adipophilin.  

Thank you for this suggestion.  The first paragraph of the discussion has been expanded and additional references cited to include findings across the spectrum of sebaceous lesions for different IHC stains.

Reviewer 2 Report

The authors reported about different expression patterns of GLUT1 in 39 sebaceous tumors, including 10 cases of sebaceous hyperplasia, too. The mentioned that GLUT1 staining differs between the different tumors. In my point of view the GLUT1 staining was related to the degree of maturation. Areas with mature sebocytes were negative for GLUT1. Therefore, the staining primarily reflects the degree of maturation, which is correlated with the tumor entity. This point was also raised up by the authors. The authors also mentioned that the distinction may not be as clear in certain cases of sebaceous adenoma where the basal proliferation is minimal…. It would be interesting to see if the staining really allows a discrimination in difficult cases. The cases shown here (Fig. 1-4), are clear cut cases, in which an additional immunostain is not needed.

Please show the same areas in HE and in GLUT1 in all figures and please give the magnification or a scale bar in each figure.  

Author Response

Author responses to reviewer comments:

Comments and Suggestions for Authors

The authors reported about different expression patterns of GLUT1 in 39 sebaceous tumors, including 10 cases of sebaceous hyperplasia, too. The mentioned that GLUT1 staining differs between the different tumors. In my point of view the GLUT1 staining was related to the degree of maturation. Areas with mature sebocytes were negative for GLUT1. Therefore, the staining primarily reflects the degree of maturation, which is correlated with the tumor entity. This point was also raised up by the authors. The authors also mentioned that the distinction may not be as clear in certain cases of sebaceous adenoma where the basal proliferation is minimal…. It would be interesting to see if the staining really allows a discrimination in difficult cases. The cases shown here (Fig. 1-4), are clear cut cases, in which an additional immunostain is not needed.

We photographed the best examples to illustrate the differences between categories of sebaceous lesions.  Additional explanation was provided at the end of the first paragraph in the discussion to better explain how GLUT1 may be helpful in difficult cases.

Please show the same areas in HE and in GLUT1 in all figures and please give the magnification or a scale bar in each figure. 

An attempt was made to identify the same areas in the tissue when the H&E and GLUT1 photographs were taken; however, since multiple stains had been performed for diagnostic purposes in most cases and archived blocks were retrieved and cut into, the difference in level in the block between the original H&E and GLUT1 stain was significant enough that the site may not appear to be the same in some of the photographs.  Original magnifications were added to the figure captions.